# Concurrent Validity and Reliability of the Sprint Force–Velocity Profile Assessed with K-AI Wearable Tech

**DOI:** 10.3390/s23198189

**Published:** 2023-09-30

**Authors:** Laurine Vantieghem-Nicolas, Jean-Benoit Morin, Thierry Cotte, Sébastien Sangnier, Jeremy Rossi

**Affiliations:** 1Université Jean Monnet Saint-Etienne, Lyon 1, Université Savoie Mont-Blanc, Laboratoire Interuniversitaire de Biologie de la Motricité, F-42023 Saint-Etienne, France; laurine.vantieghem.nicolas@etu.univ-st-etienne.fr (L.V.-N.); sebastien.sangnier@gmail.com (S.S.); jeremy.rossi@univ-st-etienne.fr (J.R.); 2ASSE, Cellule d’aide à la Performance, F-42000 Saint-Etienne, France

**Keywords:** GPS, K-Sport^®^, running, speed, force, sport performance

## Abstract

Establishing a sprint acceleration force–velocity profile is a way to assess an athlete’s sprint-specific strength and speed production capacities. It can be determined in field condition using GNSS-based (global navigation satellite system) devices. The aims of this study were to (1) assess the inter-unit and the inter-trial reliability of the force–velocity profile variables obtained with K-AI Wearable Tech devices (50 Hz), (2) assess the concurrent validity of the input variables (maximal sprint speed and acceleration time constant), and (3) assess the validity of the output variables (maximal force output, running velocity and power). Twelve subjects, including one girl, performed forty-one 30 m sprints in total, during which the running speed was measured using two GPS (global positioning system) devices placed on the upper back and a radar (Stalker^®^ Pro II Sports Radar Gun). Concurrent validity, inter-device and inter-trial reliability analyses were carried out for the input and output variables. Very strong to poor correlation (0.99 to 0.38) was observed for the different variables between the GPS and radar devices, with typical errors ranging from small to large (all < 7.6%). Inter-unit reliability was excellent to moderate depending on the variable (ICC values between 0.65 and 0.99). Finally, for the inter-trial reliability, the coefficients of variation were low to very low (all < 5.6%) for the radar and the GPS. The K-AI Wearable Tech used in this study is a concurrently valid and reliable alternative to radar for assessing a sprint acceleration force–velocity profile.

## 1. Introduction

Being able to run as fast as possible over a given distance is a key performance factor in many sports. For a 100 m sprinter, this directly defines overall performance. Sprint performance is also crucial in team sports (e.g., soccer, rugby) as this ability can be decisive for example in goal-scoring situations [1,2]. This ability is related to mechanical power (i.e., change in kinetic energy of the overall body over time) that can be produced in the horizontal direction during acceleration [3,4]. The ability to produce horizontal velocity and horizontal external force during sprint acceleration may be described by the linear sprint force–velocity relationship, which allows the force capacities expressed at different running speeds to be determined [4,5].

This relationship can be established from measurements made with laboratory tools such as force platforms [3,4,5]. In field practice, this force–velocity relationship can also be established using a simple calculation method that uses the running speed over the entire acceleration phase [3,4,5]. Using this approach, the force–velocity profile can also be measured using field tools such as a radar and, more recently, global positioning system (GPS) devices, which are now often used in sports [6,7,8].

Most elite clubs are equipped with GPS systems with technical characteristics such as sampling and analysis frequency, which can differ significantly (e.g., Catapult^®^ 10 Hz; StatSports^®^ 10 Hz; GPEXE^®^ 20 Hz; K-Sport^®^ 50 Hz, WIMU™, Fitogether^®^). This allows staff members to better quantify athletes’ physical performance and training loads [9]. However, not all GPS systems are made equal (i.e., electronic design and signal processing), and the validity and reliability of any given specific system used must be verified before the force–velocity profile can be confidently analyzed. The concurrent validity and the reliability of a GPS system are two of the most important parameters which, once verified, bring confidence about the GPS data measured [10].

Several studies, reviewed in 2016, have already tested the concurrent validity and reliability of GPS systems (various models of GPS from Catapult^®^, GPSports^®^ and WIMU™) for distance, speed and acceleration measurements [11,12], and they suggested that all the GPS devices studied in this review could accurately measure the distance covered, and had sufficient intra-unit reliability to allow multiple comparisons with the same device.

Subsequent studies have supported the use of raw GPS data to extrapolate the mechanical properties of sprint acceleration, analyzing the concurrent validity against radar, laser devices or timing gates. In 2017, Nagahara et al. [6] showed that GPS (GPEXE^®^, Exelio srl, Udine, Italy, 20 Hz and SPI proX, GPSports^®^, Canberra, Australia, 5 Hz) were not recommended for measuring sprint acceleration mechanical outputs, because although the GPEXE^®^ 20Hz had overall better results, both systems remained less reliable (mean GPS 20 Hz bias [−7.9–9.7%]; mean GPS 5 Hz bias [−5.1–2.9%]; typical error GPS 20 Hz [1.6–8.3%]; typical error GPS 5 Hz [3.9–19.2%]; correlation coefficient GPS 20 Hz [0.56–0.84]; correlation coefficient GPS 5 Hz [0.35–0.41]) than the reference devices (radar or laser).

Years later, another study showed that GPS (Apex, StatSports^®^, Dublin, Ireland, 18 Hz) could be used instead of the radar to establish a force velocity profile, as the risk of error (intra-class correlation (ICC) [0.94–0.98]; typical error [2–5.6%]) between the two GPS units was low [7].

More recently, in the study conducted by Clavel et al. [8], moderate to nearly perfect correlations (0.48–0.96) were observed between GPS force–velocity variables (Vector S7, Catapult^®^, Melbourne, Australia, 10 Hz) and the radar (Stalker Pro II^®^ Sports Radar Gun, Plano, TX, USA, 46.875 Hz). In addition, the force–velocity profile variables showed high inter-unit reliability (ICC [0.93–0.99]; typical error [0.5–2%]), demonstrating that the most recent GPS units can be used for reliably measuring force–velocity profiles. The authors concluded that this GPS type (Vector S7, Catapult^®^, Melbourne, Australia, 10 Hz) was a valid and reliable alternative to the reference method to determine force–velocity profiles [8]. Finally, a very recent study has confirmed this high inter-trial reliability, for the same GPS and radar models, provided that the same tool, the same measurements and the same analysis procedures are used [13].

The type of GPS used in the present study (K-AI Wearable Tech, KSport^®^ World Srl, Fano, Italy, 50 Hz) has never been tested for analyzing force–velocity profiles. This study has three aims: first, to assess the inter-unit and the inter-trial reliability of the force–velocity profile variables derived from K-AI Wearable Tech data. The second aim was to assess the concurrent validity of the input variables (maximal sprint speed (V_max_) and acceleration time constant (τ)). The third aim was to assess the concurrent validity of the computed output variables (theoretical maximal horizontal force (F_0_), theoretical maximal horizontal speed (V_0_) and maximal horizontal power (P_max_)) obtained with the GPS compared to the radar device. Our hypothesis was that due to its high sampling frequency, this GPS system is valid and reliable, compared with the reference device, for establishing a sprint force–velocity profile.

## 2. Materials and Methods

### 2.1. Study Design

The experimental study was conducted in a single session, lasting approximately 30 min per subject.

### 2.2. Participants

Twelve subjects including one girl (nine academic soccer players, and three physical education student/professor) participated to this study (age: 20 ± 6 years; height: 1.80 ± 0.05 m; weight: 73.7 ± 15.4 kg; body mass index: 22.8 ± 4.51 kg/m^2^). All participants were free from lower limb injuries within a period of 3 months prior to the experiment. This study was approved by the local ethics committee and was in accordance with the Declaration of Helsinki. Before giving their informed written consent, the participants were informed about the nature and the aims of the study. There were no drop-outs during the study.

### 2.3. Procedure

Participants performed a standardized warm-up, followed by two to ten sprints, depending on the participant (41 sprints in total). They performed 30 m sprints on a soccer pitch (along the midfield line), with 3 min recovery between each sprint. No specific signal was given for the start. They started from a standardized standing position and had to reach their maximum speed as quickly as possible. This 30 m distance allowed all subjects to reach their maximum speed, as the data showed that all the subjects reached a speed plateau within 30 m. The weather conditions were good, with no wind. The measurements were taken in the afternoon, and the subjects wore their own football shoes.

During these sprints, measurements were taken using two GPS (K-Sport^®^) devices named “GPS 1” and “GPS 2”, respectively, with a sampling frequency of 50 Hz and connection to four satellite constellation systems (Galileo, GPS, BeiDou and Glonass), and using a radar (Stalker^®^ Pro II Sports Radar Gun, Plano, TX, USA) with a sampling frequency of 46.875 Hz. This device was placed on a tripod (one meter above the ground) approximately 5 m behind the participants. The two GPS devices were placed on the upper back of the participants with enough space between them to ensure that the systems did not interfere with each other [14]. On average for both GPS, horizontal dilution of precision was 0.83 ± 0.11, and the average number of satellites connected during the session was 16, indicating a very good signal quality [15].

Speed and time data measured by radar and GPS were analyzed in an Excel spreadsheet based on Samozino et al.’s calculation method [3,4,5]. Processing using this Excel spreadsheet allowed us to compute the variables required to compare the two GPS units and the two measurement devices (GPS and radar). Figure 1 shows a schematic diagram of raw data processing.

To test the reliability between the two GPS units, the measurements from each device were compared. The GPS and the radar measurements were then compared to check the concurrent validity of the GPS device for computing a force–velocity profile. Finally, to test inter-trial reliability, the first two trials were compared for each participant. The data compared were the main input (V_max_ and τ) and output variables (P_max_, F_0_ and V_0_) of the model. The measured data were all processed by the same person, with the same calculation process. For each sprint, the raw running speed data were recorded via radar and GPS, and then adjusted with an exponential equation [3,4,5]. Means and standard deviations were calculated for all variables (V_max_, τ, F_0_, V_0_, P_max_).

### 2.4. Statistical Analysis

In order to make a comparison with the study of Clavel et al. [8], our statistical analysis was identical except for the comparison between radar and GPS. Indeed, in the study of Clavel et al. [8], data from the GPS units were averaged for the comparison with the radar, whereas in the present study, we have compared each GPS with the radar.

Data symmetry was verified using Fisher’s skewness coefficient. For coefficients between -1 and 1, the data were considered symmetrical [16].

GPS inter-unit reliability was assessed via the typical error of measurement (TE) expressed as a coefficient of variation (CV) and in standardized units. Reliability was also assessed via the intra-class correlation (ICC) and the smallest worthwhile change (SWC) (0.2 × between-athletes standard deviation). Using the SWC and TE, sensitivity was described as good (TE < SWC), acceptable (TE = SWC), or poor (TE > SWC). GPS inter-trial reliability was established by measuring coefficients of variation (%), the change in the mean, the standard error of measurement (SEM) and the minimal detectable change (MDC) (95% CI) for two sprints of each subject, for the various measurement tools (radar and GPS). The MDC was calculated as SEM × 1.96 × $\sqrt{2}$ [17,18]. Concurrent validity was assessed with the mean bias using the method of Bland and Altman with the typical error of estimate (TEE) in percent and standardized units, and Pearson correlation coefficients (90% confidence interval (CI)) [19].

## 3. Results

The Pearson correlation between radar and GPS 1 was moderate for τ (r = 0.49 [CI = 0.26–0.66]), and very strong for F_0_ (r = 0.84 [CI = 0.75–0.91]), P_max_ (r = 0.97 [CI = 0.95–0.98]), V_0_ (r = 0.99 [CI = 0.98–0.99]) and V_max_ (r = 0.99 [CI = 0.98–0.99]).

The Pearson correlation for radar and GPS 2 data was poor for τ (r = 0.38 [CI = 0.13–0.58]), strong for F_0_ (r = 0.79 [CI = 0.67–0.87]), and very strong for P_max_ (r = 0.95 [CI = 0.92–0.97]), V_0_ (r = 0.99 [CI = 0.98–0.99]) and V_max_ (r = 0.99 [CI = 0.99–1]).

The concurrent validity analysis results are presented in Table 1 and overall show that mean biases are small to moderate, and typical errors are small to large for the various variables.

The results of inter-unit reliability are shown in Table 2 and overall show that correlations between the two GPS are excellent to moderate, and typical errors are small to large for the different variables.

The results of inter-trial reliability are shown in Table 3 and overall show that coefficients of variation are low to very low for the radar and the GPS.

The results of the Bland and Altman analysis are illustrated in Figure 2.

## 4. Discussion

The main results of this study showed that the GPS models tested can be considered as valid in comparison to a radar and reliable for assessing the main variables of a sprint acceleration force–velocity profile.

### 4.1. Signal Quality

First, it is important to notice that during this experiment, the signal quality was good, with an average horizontal dilution of precision (0.83) lower than the one accepted for a good quality signal [14], and close to the one reported by Clavel et al. (0.74) [8] and other studies [13,20]. In addition, a large number of satellites were connected to the GPS (16 on average), which is essential for a good-quality signal [14], and also close to the data of Clavel et al. [8] and other studies [13,20]. As a result, signals from both studies were of similarly good quality, so they likely do not explain potential differences between the two studies.

### 4.2. Output Variables Validity

Strong and very strong correlations were observed between GPS and radar for the three main force–velocity profile variables (P_max_, F_0_ and V_0_). These correlations are higher than those obtained in other studies [6,8,20] (Figure 3). In our study, the GPS units tested had a sampling rate of 50 Hz, which is higher than in other studies (5 Hz, 10 Hz and 20 Hz) and closer to the radar sampling rate. Nagahara et al. compared two GPS systems with different sampling rates and showed that results were better at higher sampling rates [6].

Regarding typical errors of estimate, the values found in the present study are overall smaller or similar to those reported by Clavel et al. [8] (Figure 4) and Hoppe et al. [21]. This can also be explained by the higher sampling rate of the models tested here (50 Hz vs. 10 Hz).

However, the mean biases we obtained were greater than the mean biases obtained by Clavel et al. [8] (Figure 5), who performed measurements on an open field with no surrounding metallic structures. Contrastingly, our measurements were performed outdoors as well, but on a field surrounded by metallic barriers, which may partly explain why the force–velocity profile variables show greater mean biases in our study. In addition, the greater bias in our results could also be explained by the fact that the K-Sport^®^ manufacturer does not apply any filter to the raw data, whereas others apply a proprietary filter to the raw data. This filter, designed to improve data quality, may also help reduce bias when fitting smoothed raw data with the exponential model used. The environment can also influence measurements. In fact, we have observed a deterioration in measurements when we are close to metal structures or grandstands.

### 4.3. Input Variables Concurrent Validity

The input variables (V_max_ and τ) are directly related to the model’s output variables, which explains the observed similar levels of correlation [22] (Figure 6). Variables related to the constant speed phase (V_max_ and V_0_) determined by GPS and radar are very strongly correlated, whereas less significant correlations were obtained for parameters related to the acceleration phase (F_0_ and τ). These results are concordant with a study conducted by Buchheit et al. in 2014 [23] and also with a study conducted by Fornasier-Santos et al. in 2022 [13], which showed a good GPS accuracy at constant speed, but a poor accuracy during acceleration phases.

### 4.4. Inter-Unit Reliability

The results showed that for the various force–velocity profile variables, reliability was excellent (P_max_, V_0_ and V_max_), good (F_0_) and moderate (τ). These results were associated, respectively, with good sensitivity for P_max_, V_0_ and V_max_ and poor for F_0_ and τ (Table 2). As reported by Haugen and Buchheit in 2016 [24], sensitivity, SWC and typical error can be reduced with repeated testing. For an accurate testing, it might therefore be useful to add more sprints. Furthermore, when we compare our values with those of Clavel et al. [8], our typical errors and SWC are larger (Figure 7). This could be explained by the fact that K-Sport^®^ does not apply raw data filter, while other brands do apply raw data filter, which can improve data quality and thus reduce typical errors and SWC.

### 4.5. Inter-Trial Reliability

The inter-trial reliability was overall similar to the results of Fornasier-Santos et al. [13], i.e., low to very low coefficients of variation (all < 5.6%) and close results between radar and GPS (Table 3). These results suggest that the inter-trial reliability was good for the populations tested, and that radar and GPS can be considered as equally reliable for inter- or intra-athlete comparisons using the same device, the same measurement conditions and the same analysis procedures.

### 4.6. Limitations

Our protocol included a smaller amount of sprint data than some of the previous studies we analyzed. However, our study finally covers 72% of the speed range of the teams in the club, which enabled us to carry out this study. This is still not perfect, as we did not cover 100% of this speed range. That being said, we have no reason to think (and results from previous studies do not support a significant speed effect on concurrent validity and reliability) that different results would have been obtained if all types of maximal running speeds could have been tested. In this study, only the inter-unit reliability was tested, but it might also be interesting to study intra-unit reliability as well, as suggested by Buchheit et al. [23]. Finally, we have only assessed the concurrent validity and reliability of the GPS for force–velocity mechanical output, while it might also have been interesting to assess running distance data.

### 4.7. Practical Applications

Users of the K-AI Wearable Tech (with 50 Hz sampling frequency) may consider this tool as a valid alternative to radar for evaluating a sprint acceleration force–velocity profile.The force–velocity profile variables obtained show a good inter-unit reliability, meaning that GPS devices can be used interchangeably to measure force–velocity profiles.The force–velocity profile variables obtained show a good inter-trial reliability for both radar and GPS, but it is important to consistently use the same measurement device, either radar or GPS, to ensure valid comparisons.

## 5. Conclusions and Perspectives

This study showed that the K-AI Wearable Tech (50 Hz) is a valid and reliable alternative to radar or evaluating sprint acceleration force–velocity profile. Future studies could try to validate force–velocity profile measurements according to position on the pitch (close to barriers or stands). Or, future studies could focus on assessing the concurrent validity and reliability of this same device for establishing in situ acceleration–speed profiles based on training and game data in team sports [25].

## Figures and Tables

**Figure 1 sensors-23-08189-f001:**
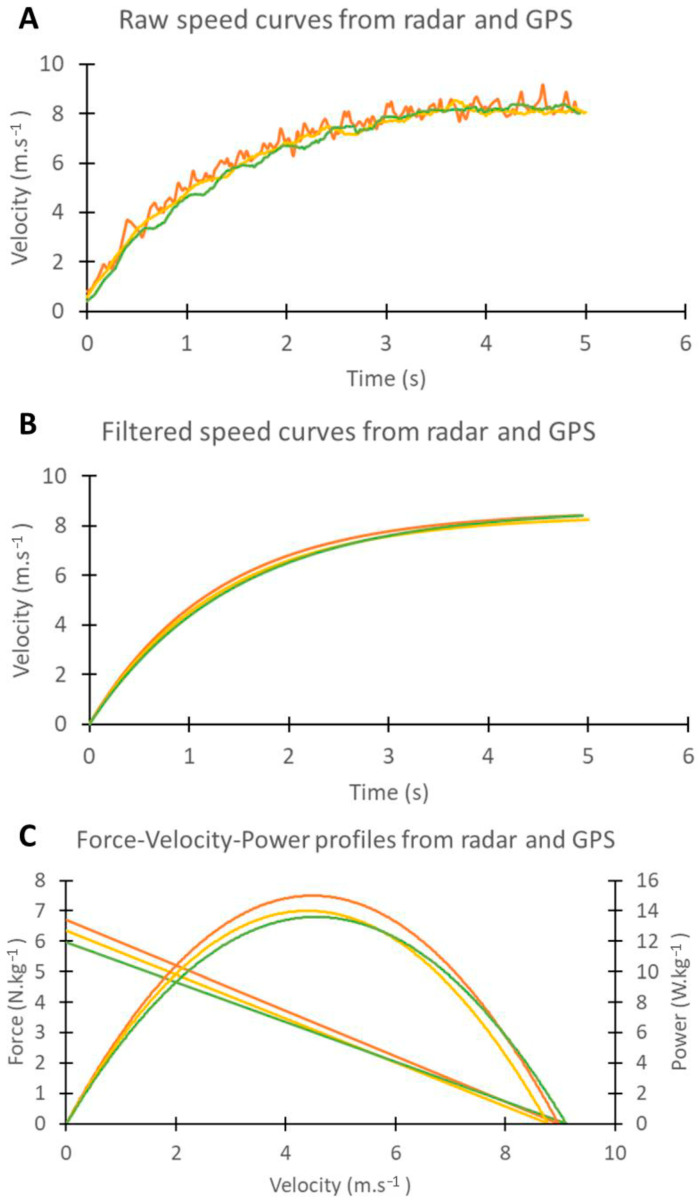
Schematic representation of the raw data processing. The upper panel (**A**) represents the raw data for a sprint. The middle panel (**B**) represents the data fitted by an exponential equation. To improve the fit, the sprint start time was set to t = 0 s using a time delay correction based on the entire signal [5]. The lower panel (**C**) shows the force–velocity–power profiles established. Red trace: radar; yellow and green trace: GPS units.

**Figure 2 sensors-23-08189-f002:**
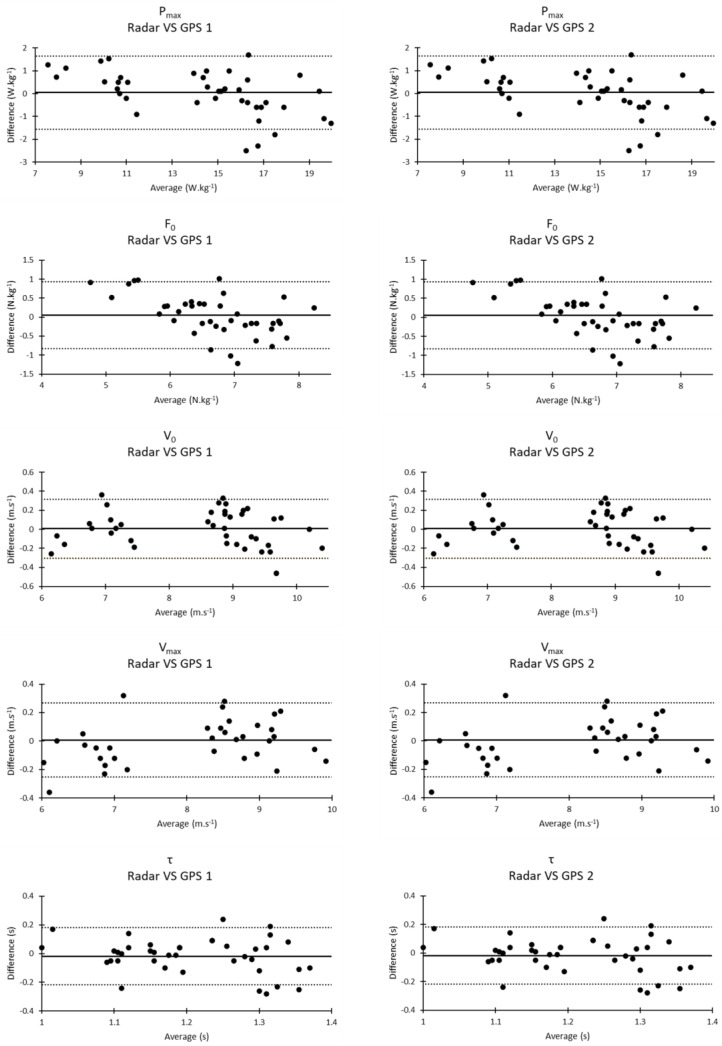
Bland and Altman comparisons between radar, GPS 1 and GPS 2. The continuous line represents the mean raw difference. The dotted lines represent the 90% confidence interval limits.

**Figure 3 sensors-23-08189-f003:**
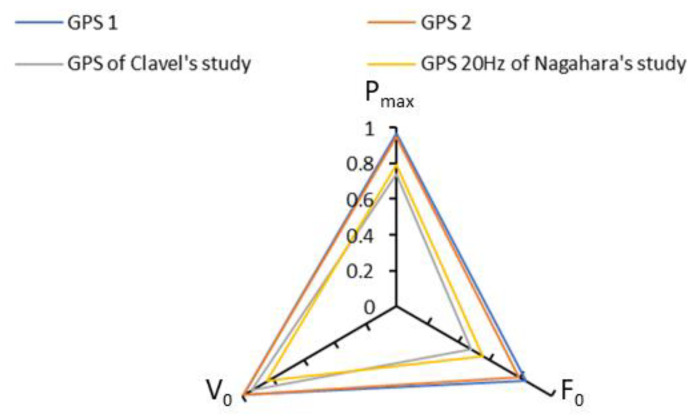
Plots showing Pearson correlation coefficients for P_max_, F_0_ and V_0_, found by our two GPS units and by GPS compared to two other studies [6,7,8].

**Figure 4 sensors-23-08189-f004:**
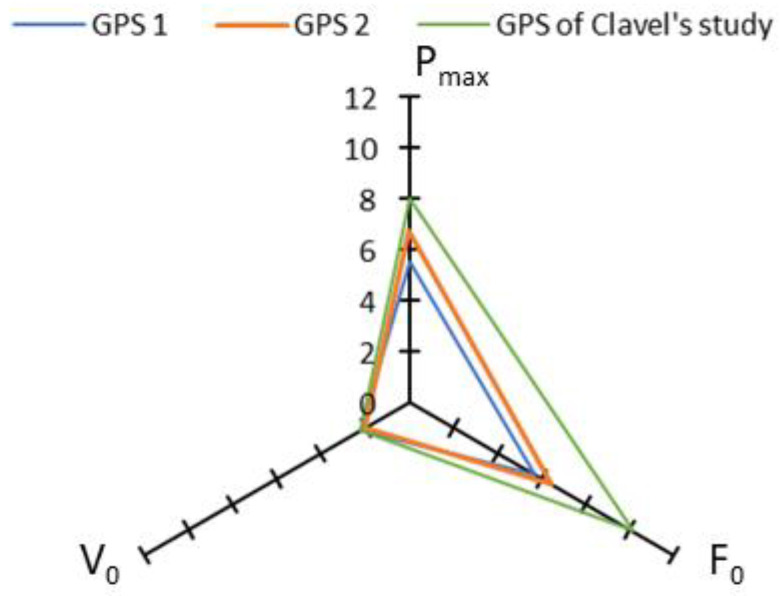
Typical errors of estimate (%) for P_max_, F_0_ and V_0_ found for the two GPS units tested here and for the GPS used in Clavel et al.’s study [8].

**Figure 5 sensors-23-08189-f005:**
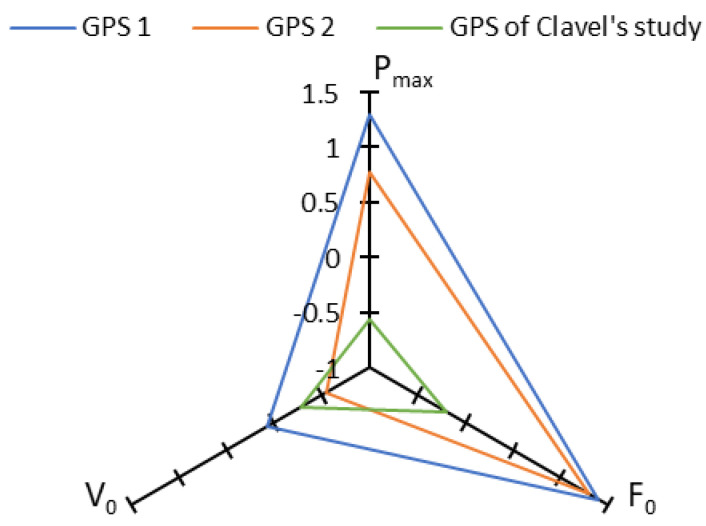
Means biases (%) for P_max_, F_0_ and V_0_, found by our two GPS units and by the GPS used in Clavel et al.’s study [8].

**Figure 6 sensors-23-08189-f006:**
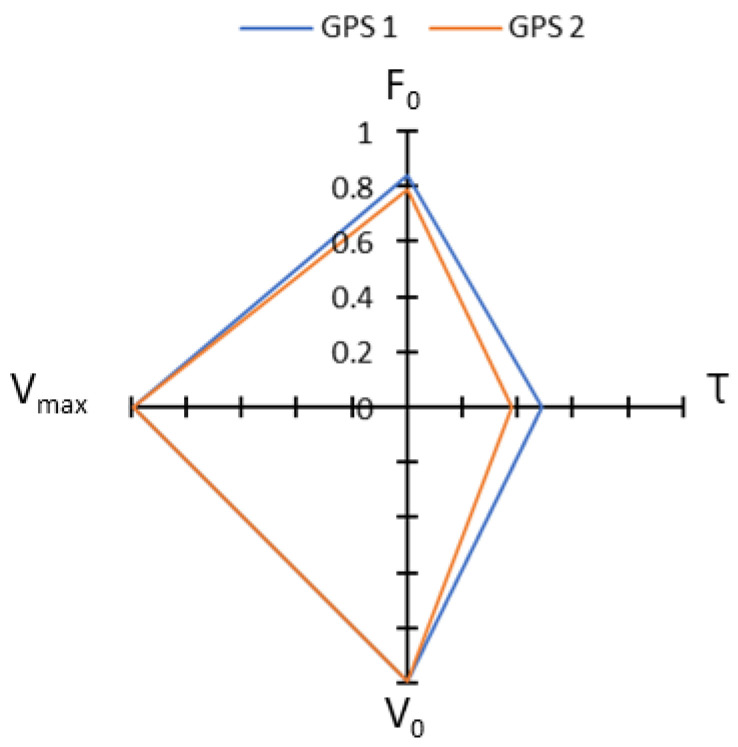
Pearson correlation coefficients between F_0_ and τ and between V_0_ and V_max_, for the two GPS devices tested.

**Figure 7 sensors-23-08189-f007:**
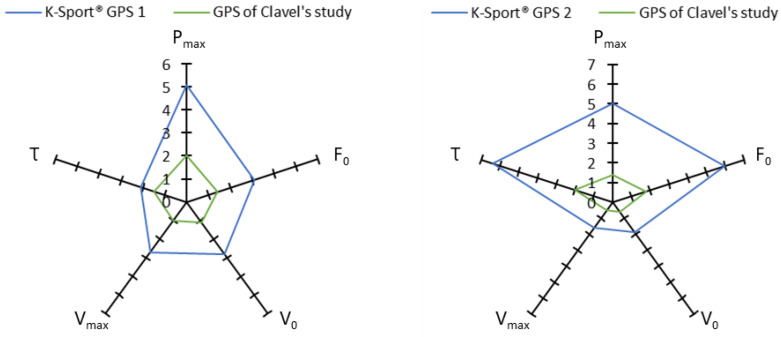
Typical errors in % (on the left panel) and SWC in % (on the right panel) for P_max_, F_0_, V_0_, V_max_ and τ, for the two GPS units tested here, and for the GPS unit tested by Clavel et al. [8].

**Table 1 sensors-23-08189-t001:** Concurrent validity analysis. Means and standard deviations [SD] of raw data are presented for the reference device (radar) and the tested devices (GPS 1 and GPS 2). CV: coefficient of variation. TEE: typical error of estimate. P_max_: maximal horizontal power. F_0_: theoretical maximal horizontal force. V_0_: theoretical maximal horizontal running velocity. V_max_: actual maximal running velocity. τ: acceleration time constant.

	P_max_ (W·kg^−1^)	F_0_ (N·kg^−1^)	V_0_ (m·s^−1^)	V_max_ (m·s^−1^)	τ (s)
GPS 1 Validity					
Radar	14.3 ± 3.1	6.7 ± 0.69	8.5 ± 1.2	8.2 ± 1.1	1.2 ± 0.1
GPS 1	14.3 ± 3.6	6.6 ± 0.97	8.5 ± 1.2	8.2 ± 1.1	1.2 ± 0.1
Mean bias (%)	1.3	1.4	0.08	0.11	−1.3
Standardized mean bias	0.01	0.08	0.00	0.01	−0.17
TEE as CV (%)	5.5	5.7	2.2	1.9	7.6
Standardized TEE	0.26	0.64	0.16	0.15	1.8
GPS 2 Validity					
Radar	14.3 ± 3.1	6.7 ± 0.69	8.5 ± 1.2	8.2 ± 1.1	1.2 ± 0.1
GPS 2	14.4 ± 3.8	6.7 ± 0.97	8.5 ± 1.2	8.2 ± 1.1	1.2 ± 0.1
Mean bias (%)	0.77	1.3	−0.54	−0.49	−1.8
Standardized mean bias	−0.02	0.05	−0.04	−0.04	−0.23
TEE as CV (%)	6.7	6.4	2	1.6	8
Standardized TEE	0.32	0.77	0.14	0.12	2.5

**Table 2 sensors-23-08189-t002:** Inter-unit reliability. Means and standard deviations (SD) for the raw data are presented for the two devices tested (GPS 1 and GPS 2). CV: coefficient of variation. TE: typical error. ICC: intra-class correlation. CI: confidence interval. SWC: smallest worthwhile change. P_max_: maximal horizontal power. F_0_: theoretical maximal horizontal force. V_0_: theoretical maximal horizontal running velocity. V_max_: actual maximal running velocity. τ: acceleration time constant.

	P_max_ (W·kg^−1^)	F_0_ (N·kg^−1^)	V_0_ (m·s^−1^)	V_max_ (m·s^−1^)	τ (s)
Reliability					
GPS 1 (mean ± SD)	14.3 ± 3.6	6.6 ± 0.97	8.5 ± 1.2	8.2 ± 1.1	**1.2** ± 0.1
GPS 2 (mean ± SD)	14.4 ± 3.8	6.7 ± 0.97	8.5 ± 1.2	8.2 ± 1.1	**1.2** ± 0.1
TE as CV (%)	5.0	6.0	1.9	1.6	6.4
Standardized TE	0.20	0.43	0.14	0.12	0.75
ICC (90% CI)	0.96 (0.94–0.98)	0.85 (0.76–0.91)	0.98 (0.97–0.99)	0.99 (0.98–0.99)	0.65 (0.47–0.78)
ICC interpretation	Excellent	Good	Excellent	Excellent	Moderate
SWC (%)	5.1	3.1	2.8	2.7	2.1
Sensitivity	Good	Poor	Good	Good	Poor

**Table 3 sensors-23-08189-t003:** Inter-trial reliability. Coefficients of variation (CV), standard deviations (SD), changes in the mean, standard errors of measurement (SEM) and minimal detectable change (MDC) were computed for each variable. P_max_: maximal horizontal power. F_0_: theoretical maximal horizontal force. V_0_: theoretical maximal horizontal running velocity. V_max_: actual maximal running velocity. τ: acceleration time constant.

	CV Inter-Trial Mean ± SD (%)			Change in the Mean ± SEM			MDC (95% CI)		
	Radar	GPS 1	GPS 2	Radar	GPS 1	GPS 2	Radar	GPS 1	GPS 2
P_max_ (W.kg^−1^)	3.1 ± 2.9	4.6 ± 4.5	4.2 ± 3.1	0.38 ± 0.94	−0.29 ± 1.4	0.2 ± 1.2	2.61	3.88	3.33
F_0_ (N.kg^−1^)	3.3 ± 2.9	5.1 ± 5	4.7 ± 3.2	0.16 ± 0.42	−0.19 ± 0.69	0 ± 0.58	1.16	1.91	1.61
V_0_ (m.s^−1^)	0.96 ± 0.7	1.4 ± 1.1	1.1 ± 0.8	0.01 ± 0.15	0.11 ± 0.22	0.07 ± 0.17	0.42	0.61	0.47
V_max_ (m.s^−1^)	0.91 ± 0.64	1.2 ± 1	1 ± 0.69	0.01 ± 0.13	0.09 ± 0.18	0.07 ± 0.14	0.36	0.50	0.39
τ (s)	3.7 ± 3.2	5.6 ± 5.7	4.9 ± 3.7	−0.02 ± 0.08	0.05 ± 0.13	0 ± 0.11	0.22	0.36	0.30

## Data Availability

Data may be provided upon detailed demand to the authors.

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
