# Peer review of "Concurrent Validity and Reliability of the Sprint Force–Velocity Profile Assessed with K-AI Wearable Tech"

_sensors, 2023, doi:10.3390/s23198189_

Round 1
Reviewer 1 Report
I would like to thank the authors for this study. It was developed properly and the results are clearly presented.
This study lets the scientific community to know the different devices that can be used to measure the sprint velocity profile, and which are validated and reliable.
A document with some comments is attached.

Author Response
Dear Editorial Board,
We thank you for allowing us to submit a revised version of our manuscript. We also thank the reviewers for their useful comments.
Please find enclosed a modified version of our manuscript that considers referees’ comments and suggestions.
As recommended, you will also find below a point-by-point response to the suggestions of the two reviewers explaining our choices on manuscript changes.
We believe that this new version is now suitable for publication in Sensors.
Yours sincerely.
Point by point response to the reviewers’ comments
Dear Editor,
Below, you will find the point-by-point response to the reviewers’ comments for the manuscript entitled « Concurrent Validity and Reliability of the Sprint Force-Velocity Profile Assessed with K-AI Wearable Tech». All comments were considered and the modifications in the manuscript have been written in red.
We would like to thank you and both reviewers for your detailed comments and helpful suggestions. We believe that the quality of the manuscript has clearly improved.
Reviewer 1
I would like to thank the authors for this study. It was developed properly and the results are clearly presented.
This study lets the scientific community to know the different devices that can be used to measure the sprint velocity profile, and which are validated and reliable.
A document with some comments is attached.
The authors present a good research paper.
- Relevance of the topic: Good.
- Introduction: Good.
- Methodology: Good.
- Results: Good.
- Discussion: Good.
However, RECONSIDER AFTER MAJOR REVISION. In general, the paper follows an adequate structure and correct scientific support and can be published considering some limitations. The study is interesting in the field of validation tests using inertial devices. However, there are a series of limitations that should be considered.
We first want to thank the reviewer for her/his valuable review that has allowed us to improve the manuscript. All comments were considered and the modifications in the manuscript have been written in red.
Specific comments.
- Title.
The title of the manuscript is correct.
- Abstract.
An introductory sentence/background needs to be added in the Abstract. Try to add more significant information related to the topic. Try to answer the following questions: “What is the
meaning of “Sprint acceleration force-velocity profile”? Why is it important to be measured?
The rest information of the abstract is very comprehensive and shows the process followed in the study.
We thank the reviewer for this comment. We have added a sentence at the beginning of the abstract:
“Establishing a sprint acceleration force-velocity profile is a way to assess an athlete's sprint-specific strength and speed production capacities.” Line 11-12
- Keywords.
Try to use different words that in the title. Please, replace “Sprint”, “Speed” and “Power”.
We thank the reviewer for this comment. The keywords "sprint", "velocity" and "power" have been replaced by "running”, “speed” and “sport performance”
- Introduction.
This section presents a coherent and clear manner with the correct support of the scientific literature. However, some specific comments are detailed below:
4.1.) Line 27-28. This sentence must be supported by a quote. Which type of sport are you referee with “track and field”? Please, specify with more detail.
We thank the reviewer for this comment. The first two sentences have been rewritten to provide more detail using the article from Haugen et al. 2019:
“Being able to run as fast as possible over a given distance is a key performance factor in many sports. For a 100-m sprinter, this directly defines overall performance. Sprint performance is also crucial in team sports (e.g. soccer, rugby) as this ability can be decisive for example in goal scoring situations”
4.2.) Line 30-35. These sentences are supported by the same quote, please, replace one of them.
We agree with the reviewer. We have replaced one of the quotes with the following article:
“A simple method for measuring power, force, velocity properties, and mechanical effectiveness in sprint running”. Samozino et al. (2016) Scand J Med Sci Sports
4.3.) Line 41-42. I think it is important to add WIMUTM, as an inertial measurement unit. It is
possible to register the data with a frequency of 100HZ, and indeed, 1000HZ. Please, add these
types of devices. Also, they have been used in different sports such as basketball, football,
rugby… and actually, Barça FC is using in each category and sport modality.
We thank the reviewer for this comment. Although we are not aware of a direct concurrent validation study for this specific technology, it has been used in research so we have mentioned this device. Line 46
4.4.) This sentence must be supported by a quote “This allows staffs to better quantify athletes’ physical performance and training loads.”
We agree with the reviewer. We have added the following quote:
Ravé, G.; Granacher, U.; Boullosa, D.; Hackney, A.C.; Zouhal, H. How to use global positioning systems (GPS) data to monitor training load in the “real world” of elite soccer. Front Physiol 2020, 11, 944. https://doi.org/10.3389/fphys.2020.00944
4.5.) Line 49-53. Please add WIMUTM devices also. Some studies used this device to assess different variables such as speed, acceleration, and power. It is important to compare and show all types of devices that we can use to measure and quantify the speed profile of the athletes.
As suggested WIMUTM device has been added. Line 55
4.6.) Line 56. “Nagahara et al.”. Remove italic style. It is not necessary. Modify in the whole
document.
We thank the reviewer for this comment. This has been modified in the manuscript.
4.7.) Line 56-62. The quote must be added next to the author. “In 2017, Nagahara et al. [5]”.
We thank the reviewer for this comment. This has been modified in the manuscript.
4.8.) Line 67-70. The same as before.
We thank the reviewer for this comment. This has been modified in the manuscript.
4.9.) The aim of the study is very clear.
We thank the reviewer for this comment.
- Methods.
Some changes must be made in this section. Please, follow the structure:
2.1. Study design
2.2. Participants
2.3. Procedure
2.4. Statistical analysis
We thank the reviewer for this comment. This has been modified in the manuscript.
5.1.) The present “design” section is not related to the design of the study. It is the procedure
carried out. Please, modify it and add a the “study design” section.
The authors must include a “design” section. To establish the design followed to carry
out the present study (theoretical study). It’s recommended to use the following methodologist:
- Montero, I.; León, O.G. A Guide for Naming Research Studies in Psychology. Int. J. Clin.
Heal. Psychol. 2007, 7, 847–862.
- Ato, M.; López-García, J.J.; Benavente, A. A Classification System for Research Designs
in Psychology. Ann. Psychol. 2013, 29, 1038–1059, doi:10.6018/analesps.29.3.178511.
We thank the reviewer for this comment. This has been modified in the manuscript. Line 96
5.2.) “Participants” section. It is possible to relocate the information in “study population”.
We thank the reviewer for this comment. Titles of sub-sections of the methodology have been renamed.
Line 92. Please, replace “body mass” with “weight”. And add the “body mass index” of
the sample.
Was there any abandonment? Do all the subjects finish the study? Please, add this
information.
We thank the reviewer for this comment. This has been modified in the manuscript. Line 101-106
None of the subjects dropped out, they all completed the study.
Did the study approve by the institution or organization? Add the number of the ethical
committee. Also, indicate that this study was carried out under the premises of the Declaration
of Helsinki (2013). This is so important, due to you are working with humans.
We thank the reviewer for this comment. This has been modified in the manuscript. Line 103-104
This study was approved by the local ethics committee and was in accordance with the Declaration of Helsinki.
5.3.) Statistical analyses. You must explain the analyses carried out in detail. All the steps
followed.
We are not sure to understand the reviewer’s comment here: all analyses were described in enough details to allow their replication and are similar to those reported in similar studies.
- Results.
The results are shown properly.
We thank the reviewer for this comment.
- Discussion.
The section Discussion is correct. Try to add more references related to the topic and based your results on previous studies. The quotes are so rare, increase them.
We thank the reviewer for this comment. New references have been added.
To our knowledge, very few papers have discussed the validation of gps to establish sprint force-velocity profile. It is for this reason that we have carried out the same statistical procedure as Clavel et al (2023) and thus compared our results.
- Conclusion.
The conclusions are correct and are related to the main objective of the study.
We thank the reviewer for this comment.
- References. The references are correct.
We thank the reviewer for this comment.
Reviewer 2 Report
The purpose of the article that I have the pleasure of reviewing was to make an assessment of inter-unit (GPS vs. radar) and inter-trial reliability of force and velocity sprinting variables, along with concurrent validity of input and computed output variables obtained via GPS device.
The main findings are that tested GPS devices (K-AI Wearable Tech) may be considered as a valid alternative to radar for measuring sprint force and velocity variables, with good inter-unit and inter-trial reliability.
The manuscript has been prepared well, but there are a few important shortcomings that require correction and clarification. Primarily, there is a lack of details regarding the conditions under which the measurements were taken, namely:
- The athletes' soccer experience level,
- 9 participants were soccer players and three others? What positions did they play? What was their level of play? Sex?
- What were the weather conditions, especially the wind? What time of day was it? What footwear was allowed? Why was the 30-meter distance chosen? In line 102, how do you ensure that this distance allowed them to reach their maximum speed?
Ln 125: „Finally, to test inter-trial reliability, two trials were compared for each participant.” Please describe each was considered for analysis if someone performed from 2 to 10 attempts. Please consider adding minimal detectable change to provide more details about the stability of measurements
Ln 11 and 17 - please explain abbreviations
Ln 16-20 - could you provide more details? participants’ detail, sprint distance, radar device,
Ln 27-28: please revise, you are using the word „performance” three times in one sentence.
Table 2 - consider adding confidence intervals for ICC
Ln 185 - maybe just „radar” to be clear instead of a „reference” device
Author Response
Dear Editorial Board,
We thank you for allowing us to submit a revised version of our manuscript. We also thank the reviewers for their useful comments.
Please find enclosed a modified version of our manuscript that considers referees’ comments and suggestions.
As recommended, you will also find below a point-by-point response to the suggestions of the two reviewers explaining our choices on manuscript changes.
We believe that this new version is now suitable for publication in Sensors.
Yours sincerely.
Point by point response to the reviewers’ comments
Reviewer 2
The purpose of the article that I have the pleasure of reviewing was to make an assessment of inter-unit (GPS vs. radar) and inter-trial reliability of force and velocity sprinting variables, along with concurrent validity of input and computed output variables obtained via GPS device.
The main findings are that tested GPS devices (K-AI Wearable Tech) may be considered as a valid alternative to radar for measuring sprint force and velocity variables, with good inter-unit and inter-trial reliability.
We first want to thank the reviewer for her/his valuable review that has allowed us to improve the manuscript. All comments were taken into account and the modifications in the manuscript have been written in red.
The manuscript has been prepared well, but there are a few important shortcomings that require correction and clarification. Primarily, there is a lack of details regarding the conditions under which the measurements were taken, namely:
The athletes' soccer experience level, 9 participants were soccer players and three others? What positions did they play? What was their level of play? Sex?
We thank the reviewer for this comment. This has been modified in the manuscript. Line
99-100
“Twelve subjects including one girl (nine academic soccer players, and three physical education student/professor) participated to this study”
What were the weather conditions, especially the wind? What time of day was it? What footwear was allowed? Why was the 30-meter distance chosen?
We thank the reviewer for this comment. This has been modified in the manuscript. Line 113-115
The distance of 30m was chosen based on existing procedures in the literature for this type of population. The test used is based on modelling data from 0 to the players maximal running speed, which is reached for this population within 30m. For example, “The influence of short sprint performance, acceleration, and deceleration mechanical properties on change of direction ability in soccer players—A cross-sectional study” Zhang et al. 2022
In line 102, how do you ensure that this distance allowed them to reach their maximum speed?
For all the tests, maximum speed was reached because we were able to observe a plateau in the speed versus time relationship. This is what the raw data consistently show, and this is what has been observed so far in the authors’ research and training practice over more than 300 cases of football or rugby players from young to elite. In our extensive experience, a very limited number of team sport players (less than 2%) reach their peak speed beyond 30-m in a linear sprint, and when it is the case, the modelling used is still valid since over 98% of top speed is reached at the 30-m mark.
Ln 125: “Finally, to test inter-trial reliability, two trials were compared for each participant.” Please describe each was considered for analysis if someone performed from 2 to 10 attempts.
The first two trials was considered for analysis. This has been modified in the manuscript. Line 139.
Please consider adding minimal detectable change to provide more details about the stability of measurements
We thank the reviewer for this comment. This has been modified in the manuscript. Table 3
Ln 11 and 17 - please explain abbreviations
We thank the reviewer for this comment. This has been modified in the manuscript. Line 13 and Line 18
Ln 16-20 - could you provide more details? participants’ detail, sprint distance, radar device,
We thank the reviewer for this comment. This has been modified in the manuscript. Line 17-19
Ln 27-28: please revise, you are using the word „performance” three times in one sentence.
We thank the reviewer for this comment. This has been modified in the manuscript. Line 29-30
Table 2 - consider adding confidence intervals for ICC
We thank the reviewer for this comment. This has been modified in the manuscript. Table 2
Ln 185 - maybe just „radar” to be clear instead of a „reference” device
We thank the reviewer for this comment. This has been modified in the manuscript. Line 201
Round 2
Reviewer 1 Report
No comments.
Reviewer 2 Report
The authors have made revisions in accordance with my feedback. At this moment, in my opinion, it meets all the criteria required for publication in the Sensors journal.